# Cathepsin B p.Gly284Val Variant in Parkinson’s Disease Pathogenesis

**DOI:** 10.3390/ijms23137086

**Published:** 2022-06-25

**Authors:** Lukasz M. Milanowski, Xu Hou, Jenny M. Bredenberg, Fabienne C. Fiesel, Liam T. Cocker, Alexandra I. Soto-Beasley, Ronald L. Walton, Audrey J. Strongosky, Ayman H. Faroqi, Maria Barcikowska, Magdalena Boczarska-Jedynak, Jaroslaw Dulski, Lyuda Fedoryshyn, Piotr Janik, Anna Potulska-Chromik, Katherine Karpinsky, Anna Krygowska-Wajs, Tim Lynch, Diana A. Olszewska, Grzegorz Opala, Aleksander Pulyk, Irena Rektorova, Yanosh Sanotsky, Joanna Siuda, Mariusz Widlak, Jaroslaw Slawek, Monika Rudzinska-Bar, Ryan Uitti, Monika Figura, Stanislaw Szlufik, Sylwia Rzonca-Niewczas, Elzbieta Podgorska, Pamela J. McLean, Dariusz Koziorowski, Owen A. Ross, Dorota Hoffman-Zacharska, Wolfdieter Springer, Zbigniew K. Wszolek

**Affiliations:** 1Department of Neurology, Mayo Clinic Florida, Jacksonville, FL 32224, USA; lmilanowski@wum.edu.pl (L.M.M.); strongosky.audrey@mayo.edu (A.J.S.); jaroslaw.dulski@gumed.edu.pl (J.D.); uitti.ryan@mayo.edu (R.U.); wszolek.zbigniew@mayo.edu (Z.K.W.); 2Department of Neuroscience, Mayo Clinic Florida, Jacksonville, FL 32224, USA; hou.xu@mayo.edu (X.H.); bredenberg.jenny@mayo.edu (J.M.B.); fiesel.fabienne@mayo.edu (F.C.F.); cocker.liam@mayo.edu (L.T.C.); beasley.alexandra@mayo.edu (A.I.S.-B.); walton.ronald@mayo.edu (R.L.W.); faroqi.ayman@mayo.edu (A.H.F.); mclean.pamela@mayo.edu (P.J.M.); ross.owen@mayo.edu (O.A.R.); 3Department of Neurology, Faculty of Health Science, Medical University of Warsaw, 02-091 Warsaw, Poland; piotr.janik@wum.edu.pl (P.J.); anna.potulska-chromik@wum.edu.pl (A.P.-C.); monika.figura@wum.edu.pl (M.F.); stanislaw.szlufik@gmail.com (S.S.); dkoziorowski@wum.edu.pl (D.K.); 4Neuroscience PhD Program, Mayo Graduate School, Mayo Clinic Florida, Jacksonville, FL 32224, USA; 5Clinical Department of Neurology, Extrapyramidal Disorders and Alzheimer’s Outpatient Clinic, Central Clinical Hospital of the Ministry of the Interior and Administration in Warsaw, 02-507 Warsaw, Poland; barcikowska@data.pl; 6Department of Neurology and Restorative Medicine, Health Institute dr Boczarska-Jedynak, 32-600 Oswiecim, Poland; m.boczarskajedynak@gmail.com; 7Department of Neurology, St. Adalbert Hospital, Copernicus PL Ltd., 80-462 Gdansk, Poland; jaroslaw.slawek@gumed.edu.pl; 8Division of Neurological and Psychiatric Nursing, Faculty of Health Sciences, Medical University of Gdansk, 80-210 Gdansk, Poland; 9Lviv Regional Clinical Hospital, 79010 Lviv, Ukraine; ljuda_fed@yahoo.com (L.F.); sanotsky@gmail.com (Y.S.); 10Uzhhorod Regional Clinical Centre of Neurosurgery and Neurology, 88018 Uzhhorod, Ukraine; cath.karpinsky@gmail.com; 11Department of Neurology, Jagiellonian University Medical College, 31-008 Krakow, Poland; krygowsk@cm-uj.krakow.pl; 12The Dublin Neurological Institute, Mater Misericordiae University Hospital, D07 W7XF Dublin, Ireland; tlynch@dni.ie (T.L.); diana.angelika.olszewska@gmail.com (D.A.O.); 13School of Medicine and Medical Science, University College Dublin, D04 V1W8 Dublin, Ireland; 14Edmond J. Safra Program in Parkinson’s Disease and the Morton and Gloria Shulman Movement Disorders Clinic, Toronto Western Hospital, Toronto, ON M5T 2S8, Canada; 15Department of Neurology, Faculty of Medical Sciences in Katowice, Medical University of Silesia, 40-055 Katowice, Poland; grzegorz.opala1@gmail.com (G.O.); asiasiuda73@gmail.com (J.S.); 16Uzhhorod National University, 88-000 Uzhhorod, Ukraine; apulyk@gmail.com; 17Applied Neuroscience Research Group, Central European Institute of Technology, CEITEC MU, Masaryk University, 601-77 Brno, Czech Republic; irena.rektorova@fnusa.cz; 18St. Anne’s University Hospital and Faculty of Medicine, Masaryk University, 601-77 Brno, Czech Republic; 19Bielanski Hospital, 01-809 Warsaw, Poland; mariusz.wid@gmail.com; 20Faculty of Medicine and Health Sciences, Andrzej Frycz Modrzewski Krakow University, 30-705 Cracow, Poland; mrudzinska@afm.edu.pl; 21Department of Medical Genetics, Institute of Mother and Child, 01-211 Warsaw, Poland; sylwia.rzonca@imid.med.pl; 22Institute of Genetics and Biotechnology, Faculty of Biology, University of Warsaw, 00-927 Warsaw, Poland; elzbieta.podgorska@imid.med.pl; 23Department of Clinical Genomics, Mayo Clinic Florida, Jacksonville, FL 32224, USA

**Keywords:** Parkinson’s disease, familial forms, monogenic forms, CTSB, fibroblasts

## Abstract

Parkinson’s disease (PD) is generally considered a sporadic disorder, but a strong genetic background is often found. The aim of this study was to identify the underlying genetic cause of PD in two affected siblings and to subsequently assess the role of mutations in Cathepsin B *(CTSB)* in susceptibility to PD. A typical PD family was identified and whole-exome sequencing was performed in two affected siblings. Variants of interest were validated using Sanger sequencing. CTSB p.Gly284Val was genotyped in 2077 PD patients and 615 unrelated healthy controls from the Czech Republic, Ireland, Poland, Ukraine, and the USA. The gene burden analysis was conducted for the *CTSB* gene in an additional 769 PD probands from Mayo Clinic Florida familial PD cohort. CTSB expression and activity in patient-derived fibroblasts and controls were evaluated by qRT-PCR, western blot, immunocytochemistry, and enzymatic assay. The CTSB p.Gly284Val candidate variant was only identified in affected family members. Functional analysis of CTSB patient-derived fibroblasts under basal conditions did not reveal overt changes in endogenous expression, subcellular localization, or enzymatic activity in the heterozygous carrier of the *CTSB* variant. The identification of the CTSB p.Gly284Val may support the hypothesis that the *CTSB* locus harbors variants with differing penetrance that can determine the disease risk.

## 1. Introduction

Parkinson’s disease (PD) is the second most common neurodegenerative disorder [1]. Clinical symptoms include bradykinesia, resting tremors, muscular rigidity, and a good response to levodopa or dopamine-agonist treatment [2]. The degeneration of dopaminergic neurons in the substantia nigra and the accumulation of α-synuclein are the main pathological hallmarks of PD [3]. It is mostly a sporadic disease, but about 10% of cases are familial [4]. Mutations in PD-related genes are usually found in patients with early-onset PD (autosomal recessive) or in patients with positive family history (autosomal dominant) [4]. In addition to the well-described causative genes, many more have been identified as potential risk factors. The most recent genome-wide association study (GWAS) revealed almost 100 susceptibility loci [5]. Although the genetics of PD in the Polish population have been explored, known genetic causes are not commonly observed [6]. This suggests that there may be specific PD-associated genetic variants within unresolved loci in the Polish population. Understanding the genetic causes of PD within families can be informative for the more frequent sporadic form of the disease. There are a number of examples in the genetics of PD that show variable penetrance at a single locus, e.g., *SNCA* and *LRRK2*, which were both identified through family-based studies but later found to harbor common variants of intermediate/low penetrance. Herein, we describe a genetic investigation of a Polish kindred with two siblings affected by PD and basic functional evaluation of the nominated *CTSB* mutant in patient-derived fibroblasts.

## 2. Results

The proband was a 58-year-old female (II-2) with 17 years history of PD (Figure 1). She developed right-hand rigidity and was responsive to levodopa treatment. After 11 years of disease onset, she developed on/off fluctuations and bilateral deep brain stimulation to the subthalamic nucleus was implanted. Her brother (Figure 1) (II-1) developed a right-hand tremor at the age of 61 and was then diagnosed with PD. After 6 years of disease onset, he continued to show a good levodopa response (Appendix A) [7]. No other known history of PD was found within the family, due to limited information from siblings who currently live in Germany.

The analysis of the three main EOPD genes—*PRKN*, *PINK1*, and *DJ1* (exons sequencing and MLPA) did not show the presence of pathogenic variants (Figure 2). The WES analysis performed for both affected subjects (II-1, II-2) identified 317 rare (MAF < 1%) missense or loss-of-function variants shared between the sibling pair. Of the 317 variants, 269 were heterozygous and 48 variants were homozygous (Appendix A). Seventeen of the identified variants, including 11 heterozygous and 6 homozygous, were previously reported to be associated with PD (Table 1).

The variant CTSB p.Gly284Val (c.851G > T) was identified as the top variant responsible for the symptoms and disease within the family. Interestingly, the *CTSB* locus was previously identified as a GWAS target, and this variant is predicted as likely pathogenic according to in silico analysis ((MutationTaster 2021, Berlin Institute of Health, Berlin, Germany), SIFT4G (Bioinformatics Institute, Singapore), CADD v1.4 (University of Washington, Hudson-Alpha Institute for Biotechnology, St. Louis, MO, USA) score 26.7) [5]. To further validate our findings, 5 independent cohorts (Table 2) were screened with no additional carriers for CTSB p.Gly284Val found. The result from gene burden analysis was negative when using the Mayo Clinic Biobank cohort and the Mayo Clinic familial PD cohort (Appendix A).

To address the potential significance of the p.Gly284Val substitution, we first inspected the surrounding sequence and the localization of the residue within the 3D structure of the protein (Figure 3). CTSB is synthesized as a preproenzyme which, upon cleavage of the N-terminal signal and pro-peptide regions, is further processed into the mature, active protease consisting of light and heavy chains (Figure 3A). Although Gly284 does not directly localize to the active site of the enzyme, it resides within a highly conserved region of the heavy chain that folds into the R-lobe of CTSB and may modulate catalytic activity (Figure 3B,C).

To functionally characterize the novel mutation, we examined mRNA and protein expressions as well as its activity and the localization using patient-derived fibroblasts from both PD patients and age-matched controls (Figure 4). During culture, there were no apparent differences in cell proliferation, morphology, or survival observed between patients and controls. Under standard, non-stress conditions, heterozygous mutant fibroblasts showed no major differences in total CTSB transcript or protein levels (Figure 4A,B). There was also no significant change in enzymatic activity of CTSB (Figure 4C). Immunocytochemistry showed similar lysosomal distribution of the CTSB signal between patients and control fibroblasts (Figure 4D).

## 3. Discussion

Our study identified a novel, rare CTSB p.Gly284Val variant in the affected family that may play a role in the pathogenesis of PD. Although we did not show differences in *CTSB* gene burden analysis in small cohorts of control and PD cases, accumulating evidence from recent large genetic studies has revealed a strong involvement of *CTSB* in PD development. The *CTSB* locus was identified as a genetic risk locus in a recent GWAS analysis [5]. *CTSB* was considered as PD expression quantitative trait loci in the transcriptomic analysis [21]. The *CTSB* variant rs1293298 was also identified as the modifier of risk and age of onset in *GBA* associated PD and Lewy body dementia [22]. On the other hand, we conducted *CTSB* gene burden analysis and CTSB p.Gly284Val genotyping which did not find any associations. In a limited number of functional studies of CTSB in PD, the direct involvement of CTSB in α-synuclein degradation was revealed. CTSB and CTSL were shown to jointly cleave α-synuclein within its amyloid region and circumvent fibril formation [13]. However, CTSB also contributed to the generation of C-terminally truncated α-synuclein species in symptomatic SNCA p.Ala53Thr transgenic mice [23]. Yet, the impact of the *CTSB* variant in PD patients on protein activity had never been investigated before. A structure-informed multiple sequence alignment of human CTSB against a selection of homologs found that Gly284 is evolutionarily conserved and lies within the heavy chain of active enzymes [24]. This indicates the potential structural importance of this residue, thus a mutation such as p.Gly284Val may adversely affect protein function.

Cathepsins are lysosomal proteases that are mainly found in the acidic compartments where they are most active, but each cathepsin has its own specific optimal pH environment [25]. Cathepsins take part in different physiological and pathological processes and play critical roles in intracellular protein degradation, energy metabolism, and immune response [26]. The importance of the lysosomal pathway in PD pathogenesis was described previously, and cathepsins belong to the most crucial lysosomal proteins [27]. Significant reduction in the lysosomal degradation capacity, substantially enlarged lysosomes, and increased lysosome number were observed in the CTSB and CTSL double knockout human neuroblastoma cells [28]. Moreover, CTSB knockout cells exhibited accumulated lysosomal protein LAMP1 (lysosomal-associated membrane protein 1) [28]. Lack of CTSB was also shown to impair lysosomal trafficking during neural development [29]. Additionally, CTSB was found to indirectly control the transcription factor EB (TFEB), which is the most important regulator of autophagy and lysosomal gene expression [30].

For functional evaluation of the variant, we did not find clear evidence supporting haploinsufficiency in the fibroblasts at basal condition. However, CTSB activity is also context-dependent, and the impact of the variant may be different in neurons or other models. Thus, a more thorough analysis of the *CTSB* variant in relevant cells of the brain and/or under disease-relevant stress conditions may be needed to uncover potential functional deficits. Lysosomal dysfunction has a well-established role in the pathogenesis of PD. Currently, there are several ongoing clinical trials targeting this pathway (for example NCT02914366, NCT04127578). Therefore, further analysis of lysosomal dysfunction in PD may result in the development of potential new therapeutic targets.

## 4. Materials and Methods

### 4.1. Clinical Examination

Four members of mixed origins, a Polish-German family (Figure 1) with typical PD from southern Poland, were recruited from the Department of Neurology in the Faculty of Health Science of the Medical University of Warsaw in Warsaw, Poland. The clinical diagnosis of PD was evaluated using the UK Parkinson’s Disease Society Brain Bank clinical diagnostic criteria at the time of specimen collection by two neurologists (DK and LM). A blood sample and a single 3 mm skin punch biopsy were collected.

### 4.2. Exome Sequencing in Sib-Pairs

Whole-exome sequencing (SureSelect Human All Exon v6 enrichment, Illumina NovaSeq 6000 platform, annotations according to Department of Medical Genetics, Institute of Mother and Child pipeline, VEP2.7) was performed on both affected family members [31]. Golden Helix SNP and Variation Suite (SVS) was used to annotate variants and identify shared variants between sib-pairs. Shared variants were filtered for coding variants excluding synonymous, potential splicing, and gnomAD European Non-Finnish minor allele frequency (MAF) < 0.01. The cosegregation of nominated variants was confirmed with Sanger sequencing in affected and nonaffected family members. The study design is summarized in Figure 2.

### 4.3. Replication Cohorts and Genotyping

Genotyping was performed in 5 different cohorts (2077 PD cases and 615 controls) collected from independent sites (Table 2). All individuals were of European Non-Finnish descent. The study was approved by the ethical review board from each institution and all participants provided informed written consent. A custom Applied Biosystems Taqman SNP Genotyping Assay (Thermo Fisher Scientific, Waltham, MA, USA) was designed for CTSB c.851G > T p.Gly284Val (NM_001908). Genotyping was performed using a QuantStudio 7 Real-Time PCR system. QuantStudio™ 7 Real-Time PCR Software (Thermo Fisher Scientific, Waltham, MA, USA) was used for analysis.

### 4.4. Gene Burden Analysis

The Mayo Clinic biobank control cohort consists of 885 unrelated samples of European Caucasian descent with no history of neurologic disease. The average age was 57 ± 15 (range 20–96) years with 438 (49.5%) males. The Mayo Clinic Florida familial PD cohort consists of 769 unrelated patients of European Caucasian descent diagnosed with PD and with a family history of PD [32]. The average age was 59 ± 18 (range 23–91) years with 462 (60%) males. All participants provided informed written consent prior to the commencement of this study. A gene burden analysis using SKAT was performed on rare (European, non-Finnish, MAF < 0.01) nonsynonymous variants with a CADD score greater than 20 within the *CTSB* gene (n = 16). Gender and age were used as covariates.

### 4.5. Protein Sequence and Structure Analysis

A multiple sequence alignment of human CTSB with a series of homologs from different species was performed in T-coffee Expresso [33,34]. The output was processed in ESPript 3 [35] to produce a structure-informed alignment. To visualize the location of Gly284, the crystal structure of human CTSB (PDB: 6AY2) [36], obtained via X-ray diffraction, was chosen for analysis in CCP4mg [37] (Figure 3).

### 4.6. Generation of Fibroblasts and Cell Culture

In an attempt to confirm the pathogenicity of CTSB p.Gly284Val, we compared the CTSB expression and activity using fibroblasts derived from patients and controls with matched age, *APOE,* and *MAPT* genotypes. Three independent experiments were performed for each functional analysis.

A piece of skin taken from the forearm under local anesthesia (size 1 mm × 1 mm) was cut and exposed to collagenase (24 h, 37 °C) (Sigma-Aldrich, St. Louis, MO, USA). After that, cells were seeded and cultured in Advance DMEM/F-12 medium (Thermo Fisher Scientific, Waltham, MA, USA) supplemented with 10% fetal bovine serum (Thermo Fisher Scientific), 1× Glutamax (Thermo Fisher Scientific), and 1× Antibiotic-Antimycotic agent (Thermo Fisher Scientific) at 37 °C in the incubator with 5.0% CO_2_. Cells were fed every 2–3 days with a fresh growth medium and split after reaching 90% confluence in proportion 1/4.

Primary human dermal fibroblasts collected from CTSB p.Gly284Val mutation carriers and controls (cryopreserved HDF cells; [Cell Applications Inc., San Diego, CA, USA, 106–05A]) were grown in Dulbecco’s modified Eagle medium (DMEM [Thermo Fisher Scientific, 11,965,118]) supplemented with 10% fetal bovine serum (FBS [Neuromics, FBS001800112]), 1% PenStrep (Thermo Fisher Scientific, 15,140,122) and 1% non-essential amino acids (Thermo Fisher Scientific, 11,140,050). All cells were grown at 37 °C, 5% CO_2_: air in the humidified atmosphere.

### 4.7. RNA Extraction and qRT-PCR

Total RNA isolation was performed using TRIzol^®^ Reagent by Life Technologies (Carlsbad, CA, USA). Frozen tissue (55 mg) was homogenized in 1 mL of TRIzol reagent using a pestle and incubated for 5 min at room temperature. The 0.2 mL of chloroform was added, the sample was capped, mixed vigorously by hand for 15 s, and incubated at room temperature for 3 min. The samples were centrifuged at 12,000× *g* for 15 min at 4 °C. Following centrifugation, the aqueous phase was transferred to a fresh tube containing 0.5 mL of isopropyl alcohol and incubated at room temperature for 10 min followed by centrifugation at 12,000× *g* for 10 min at 4 °C. The pellet was washed once with 1 mL of 75% ethanol and centrifuged at 7500× *g* for minutes at 4 °C. The RNA pellet was left to air dry briefly and reconstituted using Nuclease-Free water. RNA quality was assessed using Agilent RNA 6000 Nano kit (Agilent Technologies, Waldbronn, Germany). The Hight-Capacity cDNA Reverse Transcription Kit (Applied Biosystems, Cheshire, UK) was used to convert total RNA to single-stranded cDNA. QuantStudio™ Real-Time PCR Software (Thermo Fisher Scientific, Waltham, MA, USA) was used for the gene expression analysis (Taqman^®^ Gene Expression Assay, Probe ID Hs05518041_s1, FAM).

### 4.8. Protein Extraction and Western Blot

As previously described, cells were lysed in RIPA buffer containing protease inhibitor cocktail and phosphatase inhibitors (Sigma-Aldrich, St. Louis, MO, USA, 11697498001 and 04906837001) [38,39]. After 30 min incubation on ice, cell lysates were cleared for 15 min, 4 °C at 20,817× *g*, and protein concentrations were determined by BCA assay (Thermo Fisher Scientific, Waltham, MA, USA, 23225). Cell lysates containing 10 µg of protein were diluted in Laemmli buffer and boiled at 95 °C for 5 min before running on Tris-Glycine gels (Invitrogen, EC60485BOX). Post transfer of protein onto PVDF membranes (Millipore Sigma, St. Louis, MO, USA, IPVH00010), membranes were blocked in 5% skim milk (Genesee Scientific, El Cajon, CA, USA, 20–241) and incubated with primary antibodies against CTSB (Abcam, Cambridge, UK, 58802; 1:5000) and GAPDH (Meridian Life Sciences, Memphis, TN, USA, H86504M; 1:500,000) overnight at 4 °C. After washing, membranes were incubated in secondary antibody (Jackson ImmunoResearch, West Grove, PA, USA, 715-035-150; 1:10,000) for 1 h at room temperature. Proteins were visualized using Immobilon Western Chemiluminescent HRP Substrate (Millipore Sigma, St. Louis, MO, USA, WBKLS0500) on Pro Signal Blotting film (Genesee Scientific, El Cajon, CA, USA, 30–810 L). Quantification of western blots was performed using Image Studio Lite software. The intensity levels of protein bands were background subtracted and then normalized to the loading control.

### 4.9. CTSB Enzymatic Activity Assay

The CTSB activity was compared between patient-derived fibroblasts and control fibroblasts using a commercially available CTSB activity kit assay (Abcam, Cambridge, UK, ab65300). The harvested cell pellets were resuspended in 50 µL of chilled Cell Lysis Buffer and incubated on ice for 30 min. Then, the samples were centrifuged at 4 °C for 4 min and the supernatants were collected. The protein concentration was determined by the BCA method. To measure the CTSB activity, cell lysates containing10 ug of proteins were loaded into a 96-well plate and incubated at 37 °C with 2 µL of 10 mM CB Substrate Ac-RR-AFC (Abcam, Cambridge, UK) and 50 µL of the activity assay buffer for 2 h. Fluorescence signals in the plate were measured by a 2104 EnVision^®^ Multilabel Plate Reader (PerkinElmer, Inc., Waltham, MA, USA) at Ex/Em 400/505 nm.

### 4.10. Immunocytochemistry

As previously described, fibroblasts were seeded onto a PDL (Sigma-Aldrich, St. Louis, MO, USA, P0899) coated glass coverslips and fixed with 4% paraformaldehyde after attaching to the bottom [38,39,40]. Fibroblasts were immunostained with primary antibody against CTSB (Abcam, Cambridge, UK, 58802; 1:2000) followed by incubation with secondary antibody (Invitrogen, Waltham, MA, USA, A-11001; 1:1000) and Hoechst 33,342 (Invitrogen, Waltham, MA, USA, H21492l; 1:5000). Slices were imaged with an Axio-Observer microscope equipped with an ApoTome Imaging System (Zeiss, Oberkochen, Germany).

## Figures and Tables

**Figure 1 ijms-23-07086-f001:**
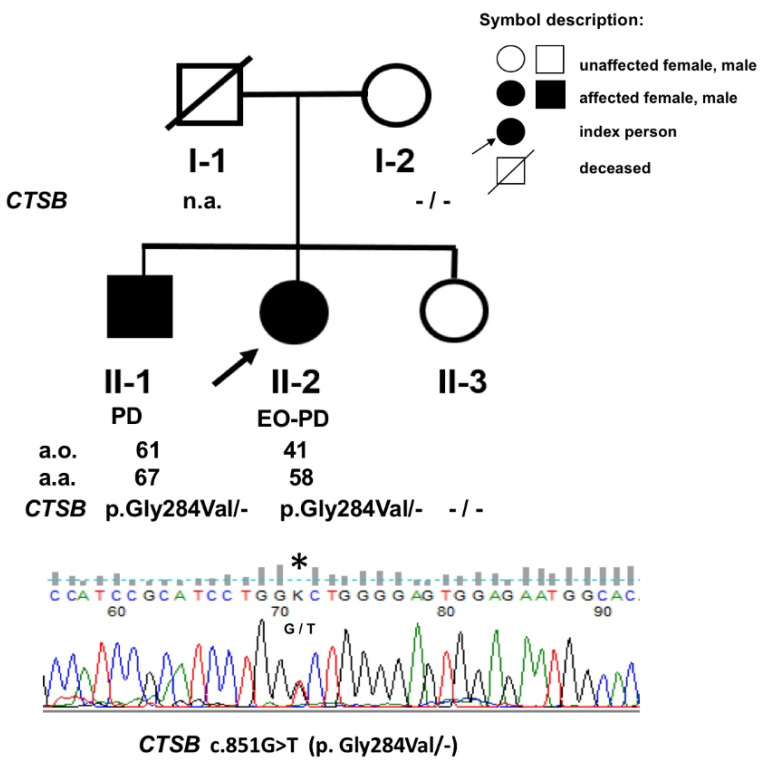
Pedigree of the Polish PD family chosen for the study. (n.a.—not available, a.o.—age of disease onset, a.a.—age of patient’s analysis, *—the mutation site).

**Figure 2 ijms-23-07086-f002:**
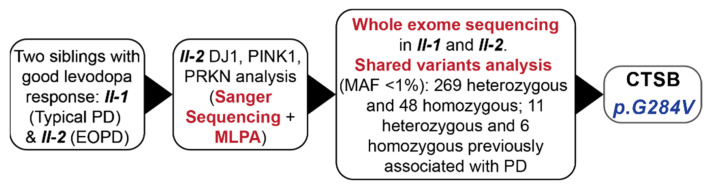
Flowchart of genetic experiments conducted on the typical PD family. EOPD = early onset Parkinson’s disease, MAF = minor allele frequency, MLPA—Multiplex ligation-dependent probe amplification.

**Figure 3 ijms-23-07086-f003:**
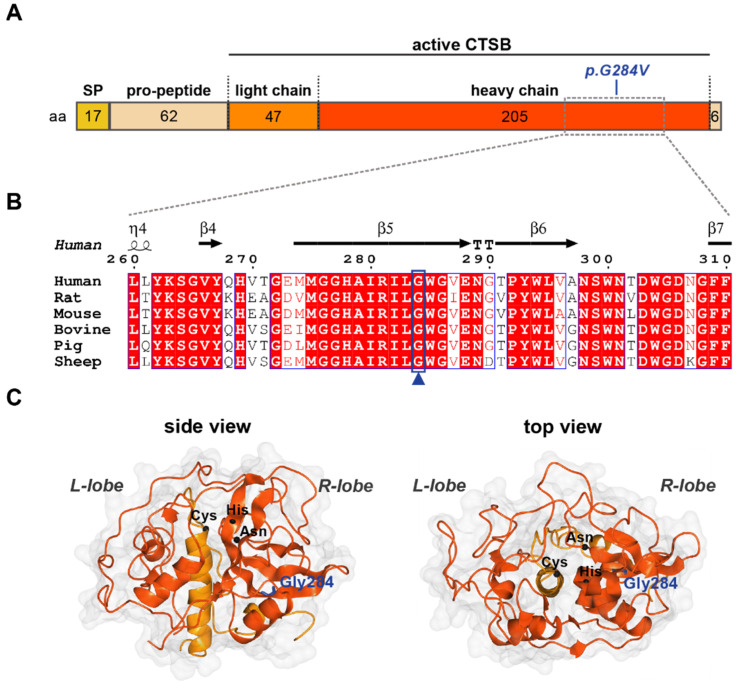
Glycine 284 is conserved in CTSB across species. (**A**) Primary structure of CTSB indicating length of fragments in amino acids (aa) and position of the p.Gly284V mutation. SP—signal peptide. (**B**) Structure-informed multiple sequence alignment of six CTSB homologs. The secondary structure for human CTSB (PDB: 6AY2) is shown above. Boxed residues are conserved: white background with red text indicates functionally equivalent residues; red background with white text indicates sequence conservation. The blue box and arrowhead highlight Gly284, which is conserved across all species analyzed. (**C**) Crystal structure of mature human CTSB (PDB: 6AY2) shown in surface/ribbon representation, indicating the left (L) and right (R) lobes which constitute the mature enzyme. Heavy and light chains are colored red-orange and orange, respectively. Gly284 (blue) is shown in cylindrical representation; C-alphas of the catalytic triad: Cys (29) His (199) and Asn (219) are depicted as black spheres.

**Figure 4 ijms-23-07086-f004:**
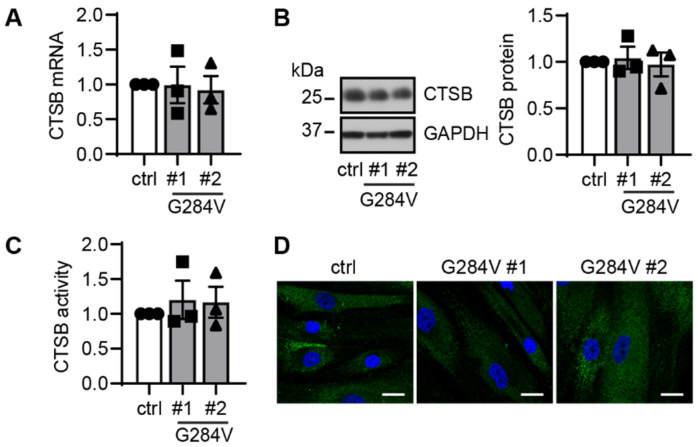
Functional analysis of CTSB in patient-derived skin cells. (**A**) qRT-PCR showed comparable mRNA levels in both CTSB mutant fibroblasts and the controls cells. (**B**) Representative western blot images of control and two CTSB mutant fibroblast lines (left). Western blot quantification showed similar levels of total CTSB protein levels in both CTSB mutant, and the WT control cells (right). (**C**) CTSB activity assay using cell lysates showed no differences in total enzymatic activity in the mutant fibroblast compared to the control. Circle—control, square—patient #1 carrying CTSB G284V, triangle—patient #2 carrying CTSB G284V. (**D**) Representative images of CTSB immunofluorescence staining (green) in fibroblasts at baseline condition. Scale bar: 20 µm. n = 3 independent experiments. Data are normalized to set control as 1 and are shown as mean with standard error. One-way ANOVA.

**Table 1 ijms-23-07086-t001:** Summary of shared variants in affected siblings previously associated with PD.

Position (GRCh38)	rs	Ref	Alt	CADD Score	Classification	Gene	Evidence
**Homozygous**							
chr6:g.32518589	rs147439581	C	T	13.6	Nonsyn SNV	HLA-DRB5	GWAS [5]
chr6:g.32519397	rs112872773	C	G,T	10.9	Nonsyn SNV	HLA-DRB5	GWAS [5]
chr6:g.32521967	rs780328684	G	TGG,-		Frameshift Del, Frameshift Sub	HLA-DRB5	GWAS [5]
chr6:g.32584262	rs150747106	C	T,G	21.9	Nonsyn SNV	HLA-DRB1	GWAS [8]
chr10:g.17849736	rs71497225	G	C	8.1	Nonsyn SNV	MRC1	Decrease the risk of PD [9]
chr10:g.17849710	rs71497223	A	G	6.7	Nonsyn SNV	MRC1	Decrease the risk of PD [9]
**Heterozygous**							
chr1:g.13782630	rs140700877	A	C	22.7	Nonsyn SNV	PRDM2	SNP more frequent observed in PD [10]
chr2:g.182756345	rs138065612	C	T	24.1	Nonsyn SNV	DNAJC10	Decrease the risk of PD [11]
chr3:g.52378778	rs201064587	G	A	21.2	Nonsyn SNV	DNAH1	SNP more frequent observed in PD [12]
chr8:g.11845732	Novel	C	A	26.7	Nonsyn SNV	CTSB	Confirmed PD/LBD GWAS loci [5,13]
chr9:g.87637944	rs200255856	G	A	32	Nonsyn SNV	DAPK1	Positive impact on LRRK2 and synuclein expression in animal models [14]
chr10:g.23104162	rs2296466	A	G	15.56	Nonsyn SNV	MSRB2	Important in regulation of mitophagy [15]
chr11:g.113387894	rs35657708	G	T	10.01	Nonsyn SNV	ANKK1	SNP more frequent observed in PD [16]
chr12:g.122340779	Novel	T	A	17.03	Nonsyn SNV	CLIP1	Increase risk of PD in LRRK2 patients [17]
chr15:g.78593232	rs76071148	T	A	15.39	Nonsyn SNV	CHRNA5	Decrease risk of PD [18]
chr16:g.1443261	Novel	C	T	36	Nonsyn SNV	CCDC154	Higher level of protein observed in serum of PD patients [19]
chr19:g.55185948	rs140748270	G	C	23	Nonsyn SNV	PTPRH	Positive association in single WES study (1000 patients) [20]

Alt—alternative, CADD—Combined Annotation Dependent Depletion, Chr—chromosome, GWAS—genome wide association studies.

**Table 2 ijms-23-07086-t002:** Basic demographic characteristic of the population genotyped for CTSB c.851G > T (p.Gly284Val) variant.

Country of Origin	PD (N Females)	Age of Onset (Mean ± SD)	Controls (N Females)
USA	997 (358)	65.2 (±11.9)	-
Poland	610 (250)	59.3 (±12.6)	248 (125)
Ireland	320 (142)	57.0 (±11.9)	343 (217)
Ukraine	122 (57)	59.9 (±11.6)	24 (12)
Czech Republic	28 (14)	59.3 (±11.8)	-

## Data Availability

Data available on request due to restrictions eg privacy or ethical. The data presented in this study are available on request from the corresponding author. The data are not publicly available due to patient’s privacy.

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
