# Peer review of "Cathepsin B p.Gly284Val Variant in Parkinson’s Disease Pathogenesis"

_ijms, 2022, doi:10.3390/ijms23137086_

Round 1
Reviewer 1 Report
This manuscript discussed the Parkinson disease pathogenesis with CTSB mutation. This is a large cooperation research. CTSB mutation is noticed because of pervious GWAS analysis. And this research further identified CTSB c.851G>T (p.Gly284Val) as the top variant. My research areas are computational and theoretical chemistry. Herein, I just provide some suggestions about the research background and the overall logic.
(1) I think this manuscript should be published on medical journal to draw more attention from medical colleague.
(2) Could you please add more background discussion about CTSB mutation. Something about the location of this mutation and the effect of this mutation on protein structure. The mutation frequency in different race.
(3) I hope there will be further mechanism research discover the link between CTSB mutation to the degeneration of dopaminergic neurons.
Reviewer 2 Report
In the manuscript, Lukasz M. Milanowski et al. identified a novel, rare CTSB p.Gly284Val variant in the affected family that may play a role in the pathogenesis of PD. This finding is important, however, they did not have enough evidences to support the function of this variant in PD.
Major concern:
- The link between this variant function with PD pathogenesis is necessary.
Minor concerns:
- In Figure 1, please label everything clear, maybe just below the circle or square. For current version, I think it is too simple and difficult to understand.
- Figure 2 is also two simple and should be improved.
- I did not find the original gels for western blot in the “Original images for Blots/gel”. This figure is the only figure they provide the functional evaluation but nothing meaningful I can get.
